# Strain topological metamaterials and revealing hidden topology in higher-order coordinates

Florian Allein [1,6], Adamantios Anastasiadis [2,6], Rajesh Chaunsali [3,6], Ian Frankel[4], Nicholas Boechler[4], Fotios K. Diakonos [5] & Georgios Theocharis [2] ✉

Topological physics has revolutionized materials science, introducing topological phases of matter in diverse settings ranging from quantum to photonic and phononic systems. Herein, we present a family of topological systems, which we term "strain topological metamaterials", whose topological properties are hidden and unveiled only under higher-order (strain) coordinate transformations. We firstly show that the canonical mass dimer, a model that can describe various settings such as electrical circuits and optics, among others, belongs to this family where strain coordinates reveal a topological nontriviality for the edge states at free boundaries. Subsequently, we introduce a mechanical analog of the Majorana-supporting Kitaev chain, which supports topological edge states for both fixed and free boundaries within the proposed framework. Thus, our findings not only extend the way topological edge states are identified, but also promote the fabrication of novel topological metamaterials in various fields, with more complex, tailored boundaries.

The field of condensed matter physics has soared to new heights with the recent discovery of topological quantum matter. Topological insulating and superconducting materials, with the ability to support robust and defect-immune manipulation of electrons[1–4], have emerged as enabling candidates for the second quantum revolution[5]. Numerous topological phenomena have also found their way from the quantum realm to the classical[6–8] despite the fundamental differences between electrons (fermions) and photons or phonons (bosons), and opened the way for new technologies relevant to optical, phononic, and mechanical computing[7,9,10], and autonomous materials[11].

An elegant combination of topological and band theory concepts relates the topological class of the bulk (infinite, periodic) system to the number of topologically robust, localized edge states on a finite sample's boundary. This connection between bulk topology and the existence of boundary states is commonly termed "Bulk-Boundary Correspondence" (BBC)[12–14]. In the standard framework, BBC is interpreted as a consequence of the fact that the symmetries necessary for a topological classification of the infinite periodic system are not broken by the boundary conditions applied to the finite system[15]. However, this approach naturally leads to a limited choice of boundary conditions for the emergence of topological boundary states. For example, in passive, finite-frequency topological mechanical metamaterials[16–18] or photonic approximations of chiral symmetric topological tight-binding models[19], only fixed boundaries have been used to establish BBC[19–22]. It is, therefore, a fundamental question whether topological states exist for other boundary conditions that break the symmetries for topological classification. If yes, could BBC

[1]Univ. Lille, CNRS, Centrale Lille, Junia, Univ. Polytechnique Hauts-de-France, UMR 8520-IEMN-Institut d'Electronique de Microélectronique et de Nanotechnologie, 59000 Lille, France. [2]Laboratoire d'Acoustique de l'Université du Mans (LAUM), UMR 6613, Institut d'Acoustique—Graduate School (IA-GS), CNRS, Le Mans Université, Le Mans, France. [3]Department of Aerospace Engineering, Indian Institute of Science, Bangalore 560012, India. [4]Department of Mechanical and Aerospace Engineering, University of California, San Diego, La Jolla, CA 92093, USA. [5]Department of Physics, University of Athens, 15784 Athens, Greece. [6]These authors contributed equally: Florian Allein, Adamantios Anastasiadis, Rajesh Chaunsali. ✉e-mail: georgios.theocharis@univ-lemans.fr

be established based on hidden symmetries? Hidden symmetries have been widely shown to exist in virtually every branch of physics[23–26]. They can be revealed after mathematical mappings, suitable coordinate transformation[27–31], or by isospectral reductions[32,33] the so-called latent symmetries.

In this article, we show that, indeed, systems with hidden chiral and particle-hole symmetries exist, and these hidden symmetries are revealed only after a choice of suitable higher-order coordinates and boundaries. Specifically, we show that finite-frequency systems with free boundaries belong to a different family of topological matter, whose topological properties become apparent not in the standard coordinates ($u$) but in higher-order (e.g., "strain") coordinates ($s$). In addition, such systems have the same level of protection against disorder as the usual passive, finite-frequency topological metamaterials. We emphasize that herein we use a mechanical model to demonstrate the possibilities enabled by the use of higher-order coordinates. Still, our results are applicable to other physical systems, too, via the appropriate mappings, as we show by an example from the photonics setting in supplementary material. Given a linear compatibility matrix $C$ defined as $s = Cu$[34], we derive equations of motion in terms of bond extensions (strains) $\ddot{s} = -D_s s$, where $D_s$ is the strain dynamical matrix. Using this, we extend the BDI class of topological mechanical metamaterials to include systems whose bulk topology is probed by the winding number of the bulk $D_{s,\text{bulk}}$ and show that BBC holds for free instead of fixed boundaries (the subscript "bulk" in $D_{s,\text{bulk}}$ denotes the infinite—or equivalently periodic—system. If there is no such subscript, we refer to the finite, bounded system with boundary conditions that break translation invariance). We call these systems strain topological metamaterials (STM). For finite-frequency metamaterials, the topological invariants can be defined for the shifted dynamical matrix[16,17]. In this work, we always imply—if not else noted—that the winding number is defined upon the shifted dynamical matrix. It is important to note that second-order ordinary differential equations, such as those describing spring-mass models, are universal models for all fields of

wave physics, from mechanical to optical[19,35], acoustical[36] systems, and electrical circuits[37], among many others.

For a complete topological characterization of a finite-frequency system, it is necessary to compare this approach to the traditional one, where topological edge states are probed via the dynamical matrix $D_u$ defined in terms of lattice displacements where $\ddot{u} = -D_u u$. For the set of fixed and free boundaries, three possibilities exist for finite-frequency topological mechanical systems, as outlined in Fig. 1. The first possibility is systems that exhibit edge states only for fixed boundaries, where their topology is encoded in the winding number of the bulk $D_{u,\text{bulk}}$. We will call this the typical case. An example of this is the mechanical Su-Schrieffer-Heeger (SSH) model[8,16,38,39] shown in Fig. 1a. The second is systems that exhibit edge states only for free boundaries, where their topology is encoded in the winding number of $D_{s,\text{bulk}}$. We use the mass dimer to demonstrate this case[40,41]. The winding number of $D_{u,\text{bulk}}$ is not well-defined in displacement coordinates, yet remarkably, $D_{s,\text{bulk}}$ restores chiral symmetry, and the value of its winding corresponds to the emergence of edge states as we show in Fig. 1b. Experiments confirm the existence of edge states for free boundaries. Finally, the third possibility is systems that exhibit edge states for both free and fixed boundaries, such that their topology is encoded in both $D_{u,\text{bulk}}$ and $D_{s,\text{bulk}}$, shown in Fig. 1c. The model we use to demonstrate this is a mechanical analog of the Kitaev chain. The Kitaev chain has drawn particular interest for its support of Majorana modes, which have been suggested as promising candidates for quantum computing[42,43]. We introduce a mechanical analog, which facilitates the topological transition between trivial and non-trivial regimes. This system exhibits edge states for both free (Fig. 1c(i)) and fixed boundaries (Fig. 1c(ii)), but for different values of its parameters. Our experimental results verify the "double BBC" predicted for this model, wherein we observe edge states for both fixed and free boundaries associated with the parameter values predicted by the winding of $D_{u,\text{bulk}}$ and $D_{s,\text{bulk}}$, respectively. While we examine one-dimensional systems belonging to the

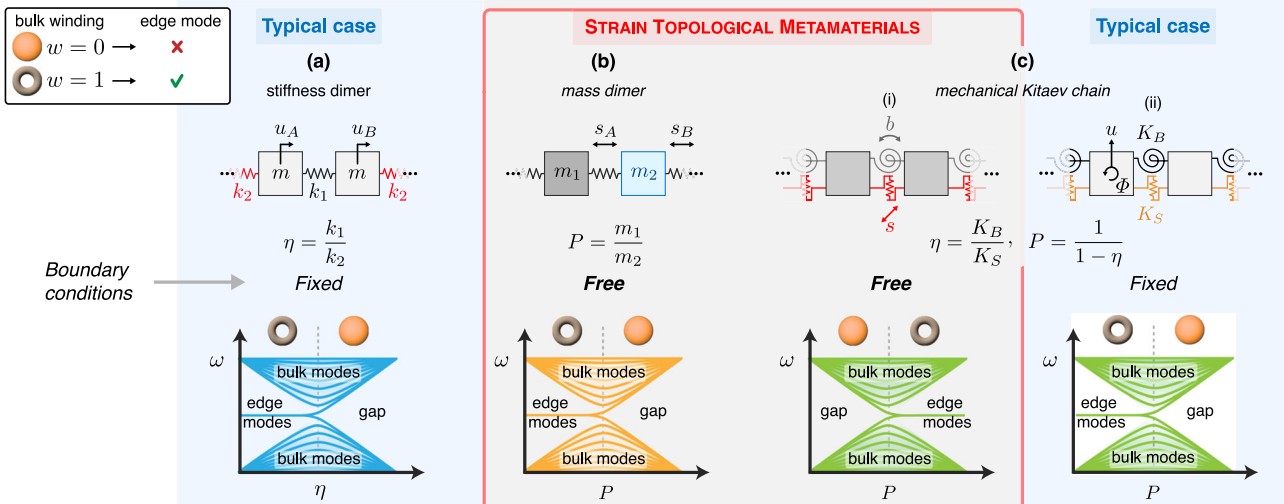

**Fig. 1 | Strain topological metamaterials.** A comparison of strain topological metamaterials (STM) to the typical case. In the top row, we present "mass-spring" schematics, which are powerful tools to model topological systems across a spectrum of physical settings, from mechanics to optics, electronic circuits, and acoustics. In the middle row, we denote the appropriate boundary conditions for the existence of BBC. In the bottom row, we show how the spectrum of each system evolves while a parameter changes adiabatically. This is compared to the predictions of the bulk winding. The orange denotes trivial winding ($w = 0$), and the doughnut non-trivial ($w = 1$). **a** The stiffness dimer can be mapped to the finite-frequency SSH model and is a typical case. Only fixed boundaries preserve the chiral symmetry of the displacement bulk dynamical matrix $D_{u,\text{bulk}}$. Edge states

appear according to the prediction of the latter's winding number. **b** The mass dimer is an STM. As a result, its chiral symmetry is revealed only in strain coordinates, and edge states can exist only for free boundaries according to the winding number of the bulk dynamical matrix in strain coordinates $D_{s,\text{bulk}}$. **c** The mechanical Kitaev chain behaves both like an STM and a typical case, depending on the applied boundaries. (i) For free boundaries, it behaves like an STM, and the winding of $D_{s,\text{bulk}}$ predicts the emergence of edge states correctly. (ii) For fixed boundaries, it behaves like a typical case, and the winding of $D_{u,\text{bulk}}$ predicts the emergence of edge states correctly. Remarkably, the topological phases of this system are interchanged when different boundaries are applied.

BDI class, other classes of strain topological metamaterials may also exist, including nonreciprocal elements and in higher spatial dimensions. Further, localized topological states are known to exist at the interface between materials of two different topological phases, which raises the possibility of new classes of interface states. Lastly, our results suggest that similar methods can be applied to other models beyond the mass-spring model[44] and to more complex, tailored boundary conditions with "higher-order" coordinates other than strain.

## Results

### Mass dimer

We begin with the mass dimer shown in Fig. 1b. This system is a periodic 1D mass-spring chain with two alternating masses, $m_1$ and $m_2$, connected with a spring of stiffness $k$. The equations of motion of the particle displacements ($u_{A|B,n}$) in the $n$th unit cell are given by:

$$m_1 \ddot{u}_{A,n} = k(u_{B,n} - u_{A,n}) - k(u_{A,n} - u_{B,n-1}) \tag{1}$$

$$m_2 \ddot{u}_{B,n} = k(u_{A,n+1} - u_{B,n}) - k(u_{B,n} - u_{A,n}), \tag{2}$$

where the first subscript denotes the sublattice within the unit cell and the second subscript $n$ denotes the unit cell number. We seek plane wave solutions of the form $\boldsymbol{\psi}_n(t) = \boldsymbol{u}(q)e^{i\Omega t - iqn}$, where $q$ is the normalized wavenumber and $\Omega$ the angular frequency. This results in the eigenvalue problem $\hat{D}_{u,\text{bulk}}(q)\hat{\boldsymbol{u}}(q) = \omega^2 \hat{\boldsymbol{u}}(q)$, where $\hat{\boldsymbol{u}} = [\sqrt{P}u_A(q), u_B(q)]^T$, $\hat{D}_{u,\text{bulk}}(q)$ is the Bloch dynamical matrix in displacement coordinates of the following form

$$\hat{D}_{u,\text{bulk}}(q) = \frac{1}{(1+P)}\begin{pmatrix} 2 & -\sqrt{P}(1+e^{-iq}) \\ -\sqrt{P}(1+e^{iq}) & 2P \end{pmatrix}, \tag{3}$$

and $\omega = \Omega/\Omega_0$ the normalized frequency with respect to the mid-gap frequency $\Omega_0^2 = k(1/m_1 + 1/m_2)$. It is well known that the edge states of the mass dimer appear for free edges when the ratio $P := m_1/m_2$ is varied[40,41]. However, their topological nature has been unknown since the dynamical matrix lacks the necessary symmetries for a topological classification. This is evident when $\hat{D}_{u,\text{bulk}}$ is written in terms of the complex Pauli matrices $\sigma_i$ ($i = x, y, z$), such that $\hat{D}_{u,\text{bulk}} = \boldsymbol{I} + d_x\sigma_x + d_y\sigma_y + d_z\sigma_z$, where $d_x = \sqrt{P}(1 + \cos q)/(1+P)$, $d_y = \sqrt{P}\sin q/(1+P)$, and $d_z = (1-P)/(1+P)$. The presence of all the $\sigma_i$ indicates that the $\hat{D}_{u,\text{bulk}}$ (up to a constant shift in the diagonal) does not anti-commute with any of these matrices. This implies the absence of chiral symmetry as in the standard SSH model[45,46].

We argue that the edge states in the mass dimer have a topological origin that can be revealed using strain coordinates. The strain coordinates for the $n$th unit cell are $s_{A,n} = u_{n,B} - u_{n,A}$ and $s_{B,n} = u_{n+1,A} - u_{n,B}$. By rearranging the equations of motion (see Methods) and assuming plane wave solutions, we arrive at the following eigenvalue problem: $D_{s,\text{bulk}}(q)\boldsymbol{s}(q) = \omega^2 \boldsymbol{s}(q)$, where $\boldsymbol{s}(q) = [s_A(q), s_B(q)]^T$, and $D_{s,\text{bulk}}(q)$ is the Bloch dynamical matrix in strain coordinates:

$$D_{s,\text{bulk}}(q) = \frac{1}{(1+P)}\begin{pmatrix} 1+P & -(P+e^{-iq}) \\ -(P+e^{iq}) & 1+P \end{pmatrix}. \tag{4}$$

The matrix $D_{s,\text{bulk}}(q)$ can be written in terms of Pauli matrices $\sigma_x$, $\sigma_y$, and $\sigma_z$, such that $D_{s,\text{bulk}}(q) = \boldsymbol{I} + d_x\sigma_x + d_y\sigma_y$ with $d_x = (P + \cos q)/(1+P)$ and $d_y = \sin q/(1+P)$. As a result, the matrix anti-commutes with $\sigma_z$ after a constant shift in the diagonal: $\sigma_z(D_{s,\text{bulk}}(q) - \boldsymbol{I})\sigma_z^{-1} = -(D_{s,\text{bulk}}(q) - \boldsymbol{I})$. In other words, the shifted $D_{s,\text{bulk}}(q)$ is chiral. Thus, the system has a well-defined winding number on the $d_x$-$d_y$ plane, as is shown in Fig. 2a. The winding number predicts a topological phase transition at $P = 1$, with $P > 1$ and $P < 1$ corresponding to trivial and non-trivial phases,

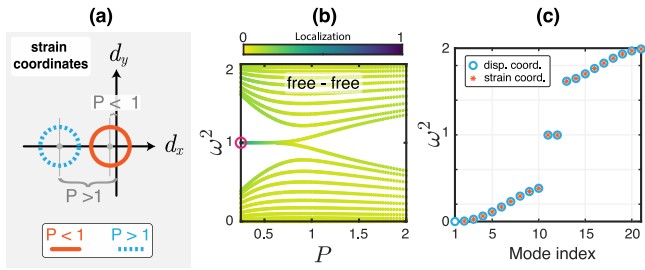

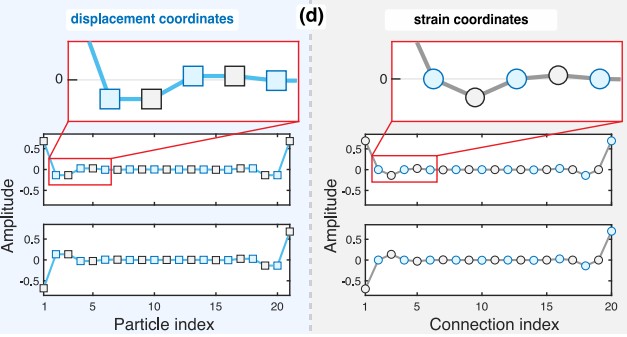

**Fig. 2 | Mass dimer. a** The chiral symmetry of mass dimer—revealed on strain coordinates—leads to a well-defined winding number. Nonzero winding makes the configurations with $P < 1$ topologically non-trivial. **b** Evolution of the spectrum of a finite chain with an odd number of particles (21) and free boundaries as we change the parameter $P$. Edge states emerge for $P < 1$. Colormap confirms localization of states inside the band gap. **c** Spectrum of a finite chain in displacement and strain coordinates at $P = 0.25$. Except for the zero mode, the spectrum is the same in both coordinates and shows chiral symmetry about the mid-gap frequency $\omega^2 = 1$. **d** Profiles of the edge states in **c**. Their chiral nature is revealed in strain coordinates.

respectively. Therefore, we expect BBC for the mass dimer—but in strain coordinates.

Figure 2b shows the spectrum of a finite chain with free boundaries and an odd number of particles (which means an even number of bonds). We witness the emergence of edge states inside the band gap for $P < 1$ (corresponding to the lighter mass on the boundaries), as expected by the strain winding number. We note that BBC dictates that the finite dynamical matrix, $D_s$, should also preserve the underlying chiral symmetry. This preservation is validated by the chiral operator for the finite matrix, which is defined as $\Gamma = \sigma_z \oplus \sigma_z \oplus \dots \oplus \sigma_z$ (see Methods).

We contrast these findings with the interpretation of the chain in the typical displacement coordinates. In displacement coordinates, the chain has a zero-frequency mode, which corresponds to the rigid body motion of the free chain and breaks chirality. The strain-description predicts all the non-zero eigenvalues of the system as is shown in Fig. 2c. We witness the chiral symmetry of these non-zero eigenfrequencies with respect to the mid-gap frequency $\omega^2 = 1$. Furthermore, in Fig. 2d, we show the profiles of the edge states at $P = 0.25$ in both displacement and strain coordinates. Once again, strain coordinates reveal the chiral nature of the chain, where the vanishing amplitude of the topological edge states at alternating bonds is akin to the mechanical SSH (stiffness dimer)[46].

Building on the idea of BBC for strain coordinates and revealing topological modes, we construct a mechanical analog of the Kitaev chain (the prototypical model for a topological superconductor) with two degrees of freedom (DOFs) per site. Specifically, particle displacement and rotation lead us to choose generalized strain coordinates involving both DOFs and probe the topological nature of the Kitaev chain. We demonstrate that this design not only obeys BBC for fixed boundaries (right column of Fig. 1c), but also shows a topological

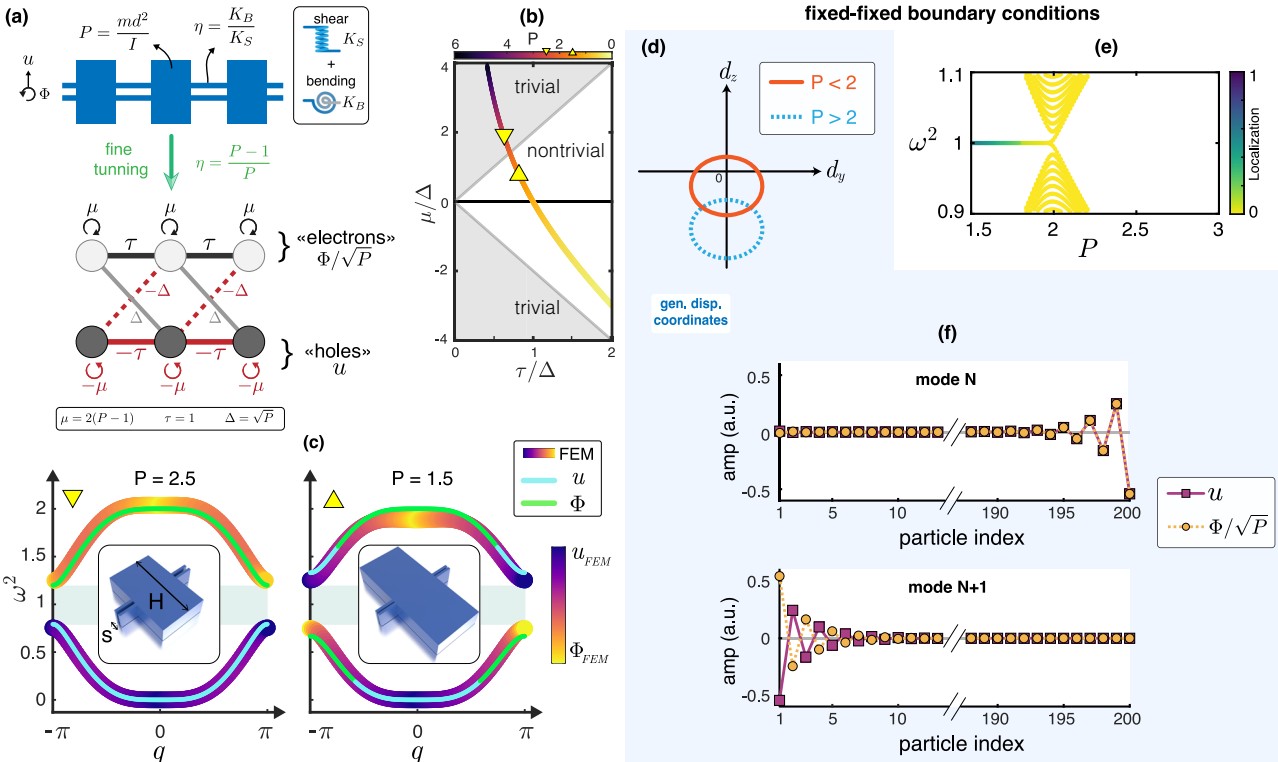

**Fig. 3 | Mechanical Kitaev chain. a** A mechanical monomer chain with transverse and rotational degrees of freedom maps to the Kitaev chain after fine-tuning. **b** Topological phase diagram of the Kitaev chain. The path of the fine-tuned mechanical chain follows the curved solid line (colorbar from yellow to black representing the value of the parameter $P$). Two cases experimentally tested herein ($P = 1.5$ and $P = 2.5$) are marked with triangles. **c** Dispersion diagrams for $P = 1.5$ and $P = 2.5$ are obtained in two ways: Analytically, via the lumped-mass model, and numerically, using the finite element method. $H$ and $s$ are the varying dimensions. Colorbar denotes modal dominance. **d** The winding number of $\tilde{D}_{u,\text{bulk}}$ suggests a topologically non-trivial phase for $P < 2$. **e** Evolution of the spectrum of finite chain with an even number of particles (200) and fixed boundaries as we change $P$. Edge states emerge for $P < 2$. **f** Profiles of the edge states in **e** for $P = 1.5$. Due to particle-hole symmetry, the profiles of effective particles $u$ and holes $\Phi/\sqrt{P}$ are either identical or differ by phase.

edge mode for free boundaries that can be explained by BBC in strain coordinates (left column of Fig. 1c).

## Mechanical Kitaev chain

In Fig. 3a, we show a mechanical structure whose dynamics are governed by two in-plane DOFs at each site (transverse displacement $u$ and rotation $\Phi$). Each site is connected with the next via two bonds corresponding to bending and shear stiffness ($K_B$ and $K_S$, respectively). We set $P = md^2/I$, the ratio of the generalized masses (particle mass $m$, lattice constant $d$, and particle mass moment of inertia $I$) and $\eta = \frac{K_B}{K_S}$, the ratio of generalized stiffnesses (with $K_B$ and $K_S$ the bending and shear stiffnesses, respectively).

In Methods and Supplementary Note 2, we analytically show that if we impose the fine-tuning: $\eta = 1 - (1/P)$, the dynamical matrix on displacement coordinates $\tilde{D}_u$ maps to a Kitaev chain[42] (the " ˜ " sign refers to the fine-tuned system). Parameter $P$ is mapped to the chemical potential $\mu$, the coupling $\tau$, and the superconducting constant $\Delta$ in the following manner: $\mu = 2(P-1)$, $\tau = 1$, and $\Delta = \sqrt{P}$. As a result, transverse displacements $u$ and the normalized rotations $\Phi/\sqrt{P}$ can be seen as particle and hole DOFs (Fig. 3a). This mapping allows us to switch between trivial and non-trivial topological phases by continuously altering the value of $P$ while retaining the fine-tuning. Since we vary $P$ in our design, we trace a 1D path in the phase space of the Kitaev chain, as is shown in Fig. 3b, wherein transitions between topologically trivial and non-trivial phases are possible. In Fig. 3c, we show the dispersion curves obtained for values of $P$ corresponding to systems in different topological phases. We observe two branches in the dispersion diagram due to the lumped-mass model having two

DOFs, i.e., $u$ and $\Phi$, per mass. We also observe that the entire spectrum ($\omega^2$) is symmetric about a mid-axis, which is $\omega^2 = 1$. This is due to the time-reversal and particle-hole symmetry, such that $\sigma_x(\tilde{D}_{u,\text{bulk}}(q) - I)\sigma_x^T = -(\tilde{D}_{u,\text{bulk}}(q) - I)$. Since $\tilde{D}_{u,\text{bulk}}(q)$ maps to the Kitaev chain BdG Hamiltonian, a finite chain with boundaries that preserve the symmetry of the bulk (i.e., a chain with fixed boundaries) will exhibit topological edge states.

Therefore, the shifted $\tilde{D}_{u,\text{bulk}}(q)$ has a well-defined winding number in the $d_y$-$d_z$ plane as shown in Fig. 3d. This suggests the existence of edge states for $P < 2$. Indeed, for a fixed chain consisting of 200 particles, two localized states emerge in the band gap for $P < 2$ as one can see in Fig. 3e. In Fig. 3f, we plot these two eigenstates, which are localized on the left and the right end of the chain. The particle-hole symmetry of the model dictates that the particle and hole DOFs of the edge mode eigenstates either exactly match (symmetric) or have opposite phases (antisymmetric)[43]. In contrast to the edge states appearing in the SSH model[39], these topologically-protected edge states have mixed polarization in terms of displacement and rotation.

We now investigate the Kitaev system in strain coordinates, which have a more complex form due to the coupling of rotational degrees of freedom with the transverse displacements. By applying the strain coordinate transformation, we obtain the bulk strain dynamical matrix $\tilde{D}_{s,\text{bulk}}(q)$, which surprisingly maps again to a Kitaev chain (as long as the fine-tuning is preserved) but with a different parameter dependence. In strain coordinates, $P$ is now replaced by $P/(P-1)$ (see Methods). In Fig. 4a, we show the winding of $\tilde{D}_{s,\text{bulk}}(q)$ predicts the inverse topological phases from those predicted by

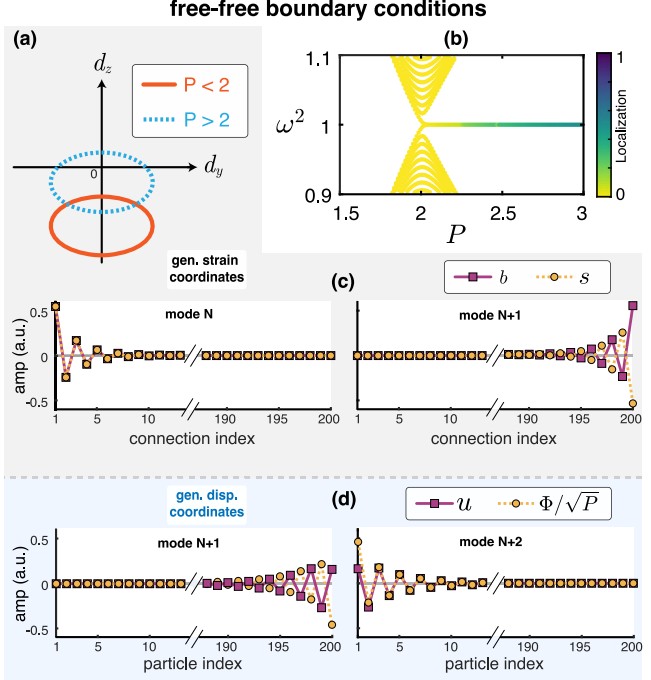

**Fig. 4 | Analysis of the Kitaev chain with free boundaries. a** The winding number of $\tilde{D}_{s,\text{bulk}}$ predicts a non-trivial phase for $P>2$ contrary to the winding of $\tilde{D}_{u,\text{bulk}}$ which predicted a non-trivial phase for $P<2$ (see Fig. 3d for comparison). **b** Spectrum of a finite chain ($N=200$) with both boundaries free as a function of $P$. Localized edge states emerge in the band gap for $P<2$. **c** Profiles of the localized states on the free edges on strain coordinates. The bending ($b$) and shear ($s$) strain coordinates follow the pattern dictated by particle-hole symmetry. **d** Profiles of the localized states on the free edges on displacement coordinates. Their profiles appear distorted.

$\tilde{D}_{u,\text{bulk}}(q)$. While a finite chain with fixed boundaries preserves particle-hole symmetry in displacement coordinates, we need a finite chain with free boundaries in order to preserve particle-hole in strain coordinates (see Supplementary Note 2 for details). As a result, we expect the emergence of edge states for a Kitaev chain with free boundaries but for the opposite parameter regimes compared to the system with fixed boundaries (topologically non-trivial regimes become trivial and vice versa).

In Fig. 4b, we show the spectrum of the Kitaev chain with free boundaries with varied $P$. Remarkably, we witness the emergence of two edge states inside the band gap for $P>2$, as predicted by the winding of the strain dynamical matrix $\tilde{D}_{s,\text{bulk}}(q)$. These hidden topological edge states exhibit the profile dictated by particle-hole symmetry when expressed in the strain coordinate system (Fig. 4c), while their form appears distorted when expressed in displacement coordinates (Fig. 4d). Building off our unique definition of generalized strain in the Kitaev chain may also open the door for establishing symmetries based on other coordinates paired with the appropriate boundaries.

## Experimental results

To verify our predictions, we prepared two test setups to experimentally probe both the mass dimer and the mechanical Kitaev chain.

For a mass-dimer chain, we expect the emergence of an edge state only at the boundary terminated by the smaller particle, as dictated by BBC. We use such a configuration for our experiments, as shown in Fig. 5a, in the form of an additively manufactured chain composed of 14 masses with free boundaries. Particle #1, located on the left of the chain, is a light mass. The mass ratio is thus to $P=0.4$ (or $P=2.5$ if considering the end with the larger mass as $m_1$). Elastic waves are

excited by striking a selected particle (#1 or #14), and the velocity of each particle is measured using a laser Doppler vibrometer. See Methods and Supplementary Note 1 for more details on fabrication, experimental setup, and data acquisition.

Figure 5b presents measured frequency response at particle #8 when the chain is excited from the left end (small mass end) or from the right side (large mass end). We observe a band gap (highlighted region). We also observe a peak inside it, but only when the particle #1 is excited, demonstrating the existence of an edge state only on the left edge, as theoretically predicted. Furthermore, we reconstruct the edge state from the experimental data in Fig. 5c. We observe excellent agreement between predictions and experiments, where amplitude decay is seen as one goes away from the left boundary.

We turn now to the experimental investigation of the mechanical Kitaev system having fixed-free boundary conditions so that both types of edge states can be observed in the system. We expect the emergence of an edge state at the fixed end, as dictated by the BBC of the fixed-fixed chain. Similarly, we expect an edge state at the free end as well, albeit for different $P$-values than the fixed chain. As such, the fixed-free chain should always have an edge state at one edge for all values of $P$ except $P=2$ (where the band gap closes). For systems with $P<2$ and $P>2$, they would support an edge state on the fixed and free end, respectively.

We additively manufacture chains of 13 masses (large cuboids) and suspend them vertically by mounting particle #1 to a fixed surface, as shown in Fig. 5d. Therefore, the system represents a fixed-free chain. We consider two chains with different $P$ and excite them with an automatic modal hammer by striking particle #2 or #13, corresponding to the fixed or free sides. Again using a laser Doppler vibrometer, we measure the velocity at multiple points along the chain. See Methods and Supplementary Note 2 for more details on fabrication, experimental setup, and data acquisition.

Figure 5e shows the experimentally measured frequency response at particle #7 when the chains with $P=2.5$ and $P=1.5$ are excited from different ends. Once again, we witness a band gap (highlighted region) and a peak inside it, which appears for a given chain and side of excitation, corresponding to the edge state. The state inside the band gap exists at the fixed end for $P=1.5$ and at the free end for $P=2.5$, as theoretically predicted.

To verify that these modes are indeed localized at different edges, we reconstruct the mode shapes from the experimental data in Fig. 5f, g. As for the mass-dimer experiments, we observe excellent agreement between predictions and experiments, where amplitude decay can be seen as one goes away from the boundaries. We also note that the edge state is localized at the free end [Fig. 5g] is different in its shape compared to its counterpart for the fixed edge, as discussed earlier, corroborating the inversion of topological phases predicted for our mechanical system.

In summary, we demonstrate theoretically and experimentally herein the existence of a different family of mechanical topological metamaterials in which bulk-boundary correspondence is realized through higher-order coordinates (e.g., strain) and suitable boundary conditions. Such topological states are also robust against several types of disorder in the system, as shown in Supplementary Note 3. We suggest that this is a general finding beyond mechanical metamaterials, as our framework can be applied to any physical system that maps to "mass-spring" systems, including, e.g., electrical or superconducting circuits, optics, and acoustics, among others. In Supplementary Note 4, we provide a supporting example via a photonic setting. Our approach enables the detection of topological edge states for boundary conditions that at first glance break topology-protecting symmetries. Furthermore, it paves the way to design topological metamaterials exploiting the interplay between boundary conditions and higher-order coordinates.

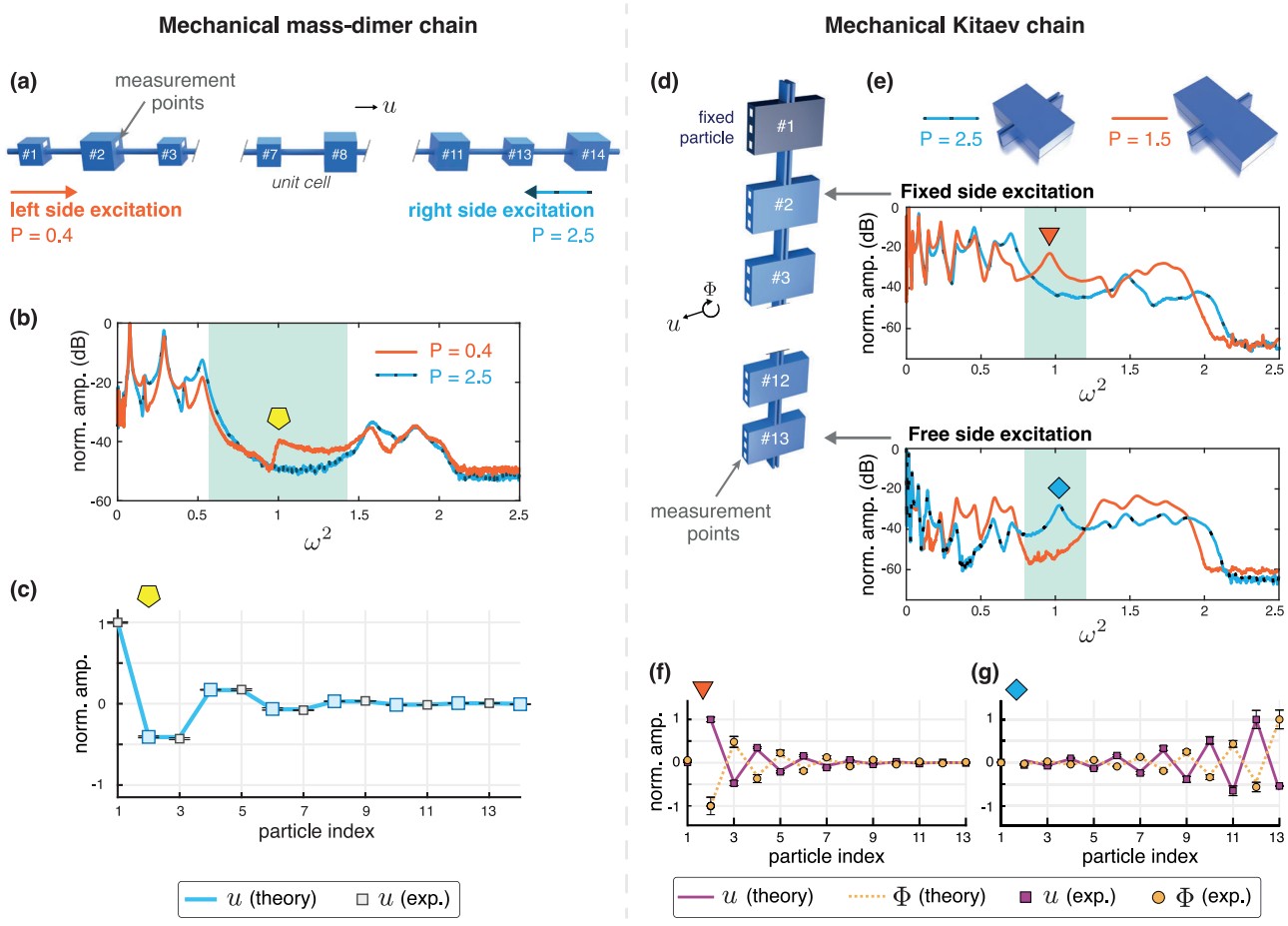

**Fig. 5 | Experimental observation of edge states in the mass dimer and mechanical Kitaev chain. a** Schematic of the experimental mass-dimer setup with free boundary conditions. **b** Measured frequency response at particle #8 when the chain is excited from the left side (#1, smaller mass) or from the right side (#14, larger mass). The shaded area corresponds to the band gap. **c** Measured amplitudes of the edge state (displayed in displacement coordinates) localized at the boundary ending with small mass and $P = 0.4$. **d** Schematic of the mechanical Kitaev chain, suspended vertically by fixing particle #1. Three points are probed on

each particle to characterize the transverse displacement and rotation. **e** Measured frequency response at particle #7 when the chains with $P = 1.5$ and $P = 2.5$ are excited at the fixed end (particle #2) or at the free end (particle #13). The shaded areas correspond to the band gap. Measured amplitudes of the edge state (displayed in displacement coordinates) localized at the fixed boundary for $P = 1.5$ (**f**), and at the free boundary for $P = 2.5$ (**g**). Error bars denote the variation (one standard deviation from averaged value) in experimental measurements.

## Methods

### Mass-dimer equations of motion on strain coordinates

We define the strain coordinates as $s_{A,n} = u_{B,n} - u_{A,n}$, $s_{B,n} = u_{A,n+1} - u_{B,n}$. We can then rearrange Eqs. (1) and (2) to get the following equations of motion in strain coordinates:

$$\ddot{s}_{A,n} = \frac{k}{m_2}(s_{B,n} - s_{A,n}) - \frac{k}{m_1}(s_{A,n} - s_{B,n-1}) \tag{5}$$

$$\ddot{s}_{B,n} = \frac{k}{m_1}(s_{A,n+1} - s_{B,n}) - \frac{k}{m_2}(s_{B,n} - s_{A,n}). \tag{6}$$

### Mass-dimer free boundaries preserve chiral symmetry on strain coordinates

A finite chain in strain coordinates, with an even number of entries, preserves chiral symmetry for the following boundary conditions: $s_{B,0} = s_{A,N+1} = 0$. In other words, the finite dynamical matrix, $D_s$, follows: $\Gamma(D_s - I)\Gamma^{-1} = -(D_s - I)$ where $\Gamma = \sigma_z \oplus \sigma_z \oplus \ldots \oplus \sigma_z$. Such boundary conditions in strain coordinates resemble a chain with free boundaries, i.e., $u_{A,N+1} - u_{B,N} = 0$. After setting $P = m_1/m_2$ and normalizing with respect to

the mid-gap frequency, the eigenvalue problem reads as follows:

$$\frac{1}{(1+P)} \underbrace{\begin{pmatrix} 1+P & -P & 0 & \ldots & & 0 \\ -P & 1+P & -1 & \ldots & & 0 \\ 0 & -1 & 1+P & -P & .. & 0 \\ \vdots & & & \ddots & & \vdots \\ 0 & \ldots & & -1 & 1+P & -P \\ 0 & \ldots & & & -P & 1+P \end{pmatrix}}_{D_{s,\text{free}}}_{2N \times 2N} \begin{pmatrix} s_{A,1} \\ s_{B,1} \\ s_{A,2} \\ \vdots \\ s_{A,N} \\ s_{B,N} \end{pmatrix} = \omega^2 \begin{pmatrix} s_{A,1} \\ s_{B,1} \\ s_{A,2} \\ \vdots \\ s_{A,N} \\ s_{B,N} \end{pmatrix}. \tag{7}$$

Clearly, strain coordinates reveal the chiral symmetry of a finite chain with free boundaries, which is absent in the displacement coordinates (see Supplementary Note 2 for more details).

### Mechanical Kitaev equations of motion of on displacement coordinates and fine-tuning

The equations of motion governing the linear dynamics of the mechanical chain considering the transverse-rotational waves are derived using the Lagrangian formalism[47,48] and are described by

the following set of differential equations:

$$m_n \ddot{u}_n = K_{S,n}(u_{n-1} - u_n) - K_{S,n+1}(u_n - u_{n+1}) \\ + dK_{S,n}(\phi_n + \phi_{n-1}) - dK_{S,n+1}(\phi_n + \phi_{n+1}), \quad (8)$$

$$I_n \ddot{\phi}_n = dK_{S,n}(u_n - u_{n-1} - d(\phi_{n-1} + \phi_n)) \\ + dK_{S,n+1}(u_{n+1} - u_n - d(\phi_n + \phi_{n+1})) \\ + d^2 K_{B,n}(\phi_{n-1} - \phi_n) - d^2 K_{B,n+1}(\phi_n - \phi_{n+1}) \quad (9)$$

where $n$ is the particle index, $m$ is the mass, $I$ is the moment of inertia, and $u_n$ and $\phi_n$ are the transverse displacement (along the $y$-axis) and rotation (around the $z$-axis), respectively, from the equilibrium position of the $n$th particle. Substituting the plane wave solutions of the form $\boldsymbol{\psi}_n(t) = \boldsymbol{v}(q)e^{i\Omega t - iqn}$ into the set of Eq. (8)–(9) leads to the eigenvalue problem:

$$D_{u,\text{bulk}}(q)\boldsymbol{v}(q) = \omega^2 \boldsymbol{v}(q), \quad (10)$$

where $\boldsymbol{v}(q) = [u(q), \Phi(q)/\sqrt{P}]^T$ is a column eigenvectors with $\Phi(q)/\sqrt{P} = d\phi(q)/\sqrt{P}$ and $P = md^2/I$. The superscript $T$ denotes the transposed vector and $D_{u,\text{bulk}}(q)$ is the dynamical matrix on displacement coordinates. $\Omega$ is the angular frequency, and $\omega = \Omega/\Omega_0$ is the normalized frequency with $\Omega_0^2 = 2PK_S/m$. The dynamical matrix $D_{u,\text{bulk}}$ can be read as:

$$D_{u,\text{bulk}}(q) = \frac{1}{2P}\begin{bmatrix} 4\sin^2(q/2) & -2i\sqrt{P}\sin(q) \\ 2i\sqrt{P}\sin(q) & 4P[\cos^2(q/2) + \eta\sin^2(q/2)] \end{bmatrix}, \quad (11)$$

where $\eta = K_B/K_S$. Under the condition:

$$\eta = 1 - (1/P), \quad (12)$$

this matrix can be fine-tuned to a new form of the dynamical matrix:

$$\tilde{D}_{u,\text{bulk}}(q) = \frac{1}{2P}\begin{bmatrix} 2P + [-2(P-1) - 2\cos(q)] & -2i\sqrt{P}\sin(q) \\ 2i\sqrt{P}\sin(q) & 2P + [2(P-1) + 2\cos(q)] \end{bmatrix}, \quad (13)$$

which resembles to the BdG Hamiltonian when taking out the constant term $2P$ of the diagonal. Comparing with the BdG Hamiltonian[43], we find that $\Delta \to \sqrt{P}, \tau \to 1, \mu \to 2(P-1)$. The matrix $\tilde{D}_{u,\text{bulk}}$ can be written in terms of the complex Pauli matrices $\sigma_x$, $\sigma_y$ and $\sigma_z$ such that:

$$\tilde{D}_{u,\text{bulk}}(q) = \boldsymbol{I} + \frac{1}{2P}\left[\left(2\sqrt{P}\sin(q)\right)\sigma_y + (-2\cos(q) - 2(P-1))\sigma_z\right]. \quad (14)$$

## Mechanical Kitaev equations of motion on strain coordinates

We are interested in the in-plane degrees of freedom that are decoupled from the longitudinal displacements. Strains are related to the transverse displacement $u_n$ and rotation $\phi_n$ DOFs in the following manner:

$$s_n = u_{n+1} - u_n - d(\phi_{n+1} + \phi_n) \quad (15)$$

$$b_n = d(\phi_{n+1} - \phi_n), \quad (16)$$

where $s_n$ and $b_n$ stand for the $n$th shear and bending strain, respectively. To express the dynamical matrix in these coordinates, we utilize Newton's equation for transverse displacements and rotations:

$$m\ddot{u}_n = K_S s_n - K_S s_{n-1} \quad (17)$$

$$I\ddot{\phi}_n = d(K_S s_n + K_S s_{n-1} + K_B b_n - K_B b_{n-1}). \quad (18)$$

We take the second derivative with respect to time in Eqs. (15) and (16) and substitute the corresponding expressions from Eqs. (17) and (18). The resulting dynamical equations for the strains are:

$$\ddot{s}_n = \frac{K_S}{m}\left(s_{n+1} + s_{n-1} - 2s_n - \frac{md^2}{I}(s_{n+1} + s_{n-1} + 2s_n) \\ - \frac{md^2}{I}\frac{K_B}{K_S}(b_{n+1} - b_{n-1})\right), \quad (19)$$

$$\ddot{b}_n = \frac{K_S}{m}\left(\frac{md^2}{I}(s_{n+1} - s_{n-1}) \\ + \frac{md^2}{I}\frac{K_B}{K_S}(b_{n+1} + b_{n-1} - 2b_n)\right). \quad (20)$$

We then define: $\boldsymbol{s}(\boldsymbol{q}) = [s(q), b(q)/\sqrt{\frac{P}{P-1}}]^T$, and assume solutions of the form $\boldsymbol{\psi}_n(t) = \boldsymbol{s}(q)e^{i(\Omega t - qn)}$ to arrive at the eigenvalue problem:

$$D_{s,\text{bulk}}(q)\boldsymbol{s}(q) = \omega^2 \boldsymbol{s}(q), \quad (21)$$

where $D_{s,\text{bulk}}(q)$ denotes the "strain" dynamical matrix and $\omega$ is again the normalized frequency $\Omega/\Omega_0$ with $\Omega_0^2 = 2PK_S/m$. Since we keep the constraint $\eta = 1 - 1/P$, the strain dynamical matrix reads as:

$$\tilde{D}_{s,\text{bulk}}(q) = \frac{P-1}{2P}\left(\begin{array}{cc} \frac{2P}{P-1} + \frac{2}{P-1} + 2\cos(q) & -2i\sqrt{\frac{P}{P-1}}\sin(q) \\ 2i\sqrt{\frac{P}{P-1}}\sin(q) & \frac{2P}{P-1} - \frac{2}{P-1} - 2\cos(q) \end{array}\right). \quad (22)$$

We verify that apart from a constant shift $\frac{2P}{P-1}$ in the diagonal, the matrix possesses particle-hole symmetry.

## Localization index calculation

Figures 2b, 3e, and 4b display a colormap (from yellow to blue), which represents a localization index. For an eigenvector with $N$ masses, we use the localization index (i.e., inverse participation ratio) defined as:

$$\text{IPR} = \frac{\sum_{n=1}^N A_n^4}{\left(\sum_{n=1}^N A_n^2\right)^2}, \quad (23)$$

where $A_n = \sqrt{u_n^2}$ or $A_n = \sqrt{u_n^2 + \frac{\Phi_n^2}{P}}$ for the mass dimer and Kitaev analyses, respectively. The inverse participation ratio (IPR) is equal to zero (yellow color) when all particles are moving (delocalized mode) and is near to one (blue color) when only a few particles are moving compared to the total length of the chain as in the case of a spatially localized mode.

## Finite element method simulations

Numerical dispersion curves are calculated via FEM using COMSOL Multiphysics software, which is presented in Fig. 3c. These results are computed by modeling the unit cells (shown in the insets of Fig. 3c) in three dimensions and applying periodic boundary conditions on the sides of the two parallel beams. The following mechanical parameters are used for the material adopting a linear elastic constitutive law: density $\rho = 1180$ kg/m³, Young's modulus $E = 2.74$ GPa, and Poisson ratio $\nu = 0.38$. Three-dimensional domains are meshed by means of three-dimensional 8-node hexahedral quadratic elements of maximum size $L_{FE} = 0.5$ mm, which is found to provide accurate eigensolutions up to the frequency of interest. The colormap reported in Fig. 3c describes the dominant component of the motion of the mass (such as $\frac{|u_f|}{|u_f| + |\phi_f|}$, where the index $f$ denotes the mode index). It varies from pure rotation (yellow) to pure transverse displacement (dark blue).

### Fabrication

The experimental samples are fabricated through additive manufacturing (Stratasys Objet350 Connex3). Thermoplastic polymer (VERO™), with the following nominal properties: density $\rho = 1180$ kg/m$^3$, Young's modulus $E = 2.74$ GPa, and Poisson ratio $\nu = 0.38$.

### Mass-dimer experiments

The mass-dimer sample is composed of 14 masses, having a total length of 35.7 cm. The masses are in the shape of a cube of sides 8 mm and 10.8 mm for the "small" and "large" masses, respectively. The beams connecting the masses are cylinders of length 15 mm and radius 1 mm. The complete design of the mass-dimer sample is presented in Supplementary Note 1. The sample is suspended on thin nylon strings to achieve free boundary conditions for all the particles. The velocity of each particle is measured on one point on each cube and averaged over 20 repetitions. The laser sensitivity was set to 100 mm/s/V.

### Mechanical Kitaev chain experiments

The specimens consist of two classes of chains hosting 13 masses for a total length of 32.2 cm. The complete design of the fine-tuned manufactured samples is presented in Supplementary Note 2. The experimental wave velocities are measured (and averaged over 10 repetitions) at three points on each particle to decompose the translational and rotational motions. The laser sensitivity was set to 50 mm/s/V. The experimentally measured spectra for all particles and additional measurements are presented in Supplementary Note 2.

## Data availability

The data generated in this study are provided in the manuscript and Supplementary Information. All other data that support the plots within this paper and other findings of this study are available from the corresponding authors upon request.

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

## Acknowledgements
We thank I. Kiorpelidis for earlier investigation on the mass-dimer system. We also thank N. Herard, B. Skoropys, and M. Coimbra for earlier investigation on the experimental realization of a mass-spring system via additive manufacturing. R.C. acknowledges the funding support by the Science and Engineering Research Board (SERB), India, through the Start-up Research Grant SRG/2022/001662. I.F. acknowledges support from the NDSEG Fellowship Program through the US Army Research Office. N.B. acknowledges support from the US Army Research Office (Grant No. W911NF-20-2-0182). A.A. acknowledges support from the Greek State Scholarships Foundation (I.K.Y) as part of the Nikolaos D. Xrysovergis grant.

## Author contributions
F.A., A.A. and R.C. contributed equally to this work. A.A. conceived the strain description and developed the theoretical framework under F.D. and G.T.'s supervision. R.C. analyzed the fine-tuning for the mechanical Kitaev chain and developed the theory on displacement coordinates. F.A., I.F. and N.B. conceived the general beam modeling. F.A. designed the experimental samples, performed the experiments, and analyzed the data. All authors contributed to the conceptualization of the project and the writing of the manuscript.

## Competing interests
The authors declare no competing interests.
