## [Peer Review File · Nature Communications]

Strain topological metamaterials and revealing hidden topology in higher-order coordinatesREVIEWER COMMENTS

Reviewer #1 (Remarks to the Author):

In this work, the authors have demonstrated a strain elastic metamaterial with hidden chiral and particle-hole symmetries that supports topological edge states for tailored boundaries (fixed or free boundaries). This is a timely topic and the authors provide a novel approach that unveils topological properties under higher-order (strain) coordinate transformations and choice of boundary. The experimental demonstrations are good and clear. It is a solid and valuable piece of work that in my opinion deserves to be published in Nature Communications. However, I think there are still some issues (listed below) need to be addressed before recommending publications.

1.The words “hitherto unknown” and “previously undiscovered” in the abstract seem a bit over the top. As I stated above this is a solid and scholarly piece work that does not need these embellishments. Such expressions may come across also as rather subjective and a bit slogan like.

2.The authors claim that the hidden chiral and particle-hole symmetries are related to the coordinate transformations and boundary conditions in their topological system. The authors should clarify more on this issue in the text and incorporate some corresponding parts of the Supplementary Material (SM) into the main text. This is a crucial point that needs to be emphasized.

3.I wonder whether the topological elastic waves system has defect immunity. For instance, I wonder if the different boundary conditions could affect the robustness of topologically protected edge states. Can the authors provide corresponding verification?

4.The full names of these abbreviations “BDI”, “ODEs” and “DOFs” should be given at the first use.

5.It is suggested that the authors make a summary at the end of the result section of the text. This can help general readers appreciate the importance of their work.

Reviewer #2 (Remarks to the Author):

In this work, the authors propose an interesting family of mechanical systems named “strain topological metamaterials”. These metamaterials support topological edge states protected by hidden symmetries, that is, symmetries that are not explicitly presented in the original Hamiltonian. The authors show two models (the mass dimer and the mechanical Kitaev models) with such intriguing behavior and address the important role of boundary conditions in preserving the symmetry. Overall, I think this work is of high quality and its results could be impactful in the whole topological classical wave community. I think that this article is suitable for Nature Communications, provided that the following issues could be satisfactorily addressed.

1. The authors use the name “strain topological metamaterials” in the title. However, it is not clear what the physical meaning of “strain” is, and it seems that D_s and D_u are just related by a basis transformation. Usually, people would expect that strain will change the positions of atomic sites (but not the basis), thus modifying the coupling terms in a Hamiltonian. But here it looks that the on-site terms are also changed. It would be beneficial if the authors can plot the original and strain coordinates and visualize how strain is applied.
2. There is another type of hidden symmetry (also call latent symmetry) that refers to the symmetry of a sub/reduced part of a system (like in Ref. [26]). I think the authors should mention this to avoid potential conceptual misleading.
3. An important yet unaddressed question is how the hidden symmetry can protect the topological states in the real system. Using the chiral symmetry as an example. For a conventional topological system with chiral symmetry, symmetry-preserving disordering (e.g., disorder in the couplings) cannot move the topological edge states from zero energy. However, for the hidden chiral symmetry, it is not clear how it can be preserved upon perturbation/disorder to the real system. From Eq. (3), it seems that any perturbation/disorder in P will destroy the chiral symmetry.
4. There are two models in the paper. But only one of them is experimentally demonstrated (i.e., the Kitaev model). In my opinion, the mass dimer model is the minimal model to

demonstrate the authors' idea (and it is easier to realize in experiments). Thus, I suggest the authors also put some experimental results in Fig. 2.

5. How is the quantity "localization" calculated? (e.g., in Fig. 2(b))

6. It would be good for understanding if a colormap can be added to the yellow curve in Fig. 3(b) to show the values of P on the curve.

Reviewer #3 (Remarks to the Author):

The authors propose the concept of "strain topological metamaterials", in which the topology is revealed only in strain coordinates, and the edge states can exist only for free boundaries. The concept proposed in the paper is interesting. However, I cannot recommend the publication of the work for several reasons:

1- The paper is too technical. I found it more suitable for a physics journal like Physical Reviews. The focus on the experimental side is very little, restricted to a very specialized case of mass spring system. If the concept was demonstrated in photonics, I would likely recommend the publication of the work. However, in the current form, the real applicability of the proposed theory is questionable to me. The authors state that the second order ODEs, such as those describing spring-mass models, are universal models for all fields of wave physics, from mechanical to optical [33], acoustical [34] systems. This statement, however, is not enough in my eye to claim that the proposed concept is easily extendable to optics or acoustics. That being said, as long as the experimental realization of the model is limited to this simple mass spring experiment, I cannot recommend the publication of the work.

2- The authors also do not show the applicability of the proposed concept in higher dimensions. A typical TI is often used for robust transportation of waves. But I am afraid that the proposed concept will lead to a novel form of topologically protected transport. Can the authors show the robust transport of energy in the strained coordinates? In my

opinion, it is not feasible. If it is true, the applicability of the proposed concept is mitigated more.

Responses to the reports of Referees – NCOMMS-23-06225-T

Reviewer #1

In this work, the authors have demonstrated a strain elastic metamaterial with hidden chiral and particle-hole symmetries that supports topological edge states for tailored boundaries (fixed or free boundaries). This is a timely topic and the authors provide a novel approach that unveils topological properties under higher-order (strain) coordinate transformations and choice of boundary. The experimental demonstrations are good and clear. It is a solid and valuable piece of work that in my opinion deserves to be published in Nature Communications. However, I think there are still some issues (listed below) need to be addressed before recommending publications.

RESPONSE: We thank the reviewer for appreciating the quality and novelty of this work.

1. *The words “hitherto unknown” and “previously undiscovered” in the abstract seem a bit over the top. As I stated above this is a solid and scholarly piece work that does not need these embellishments. Such expressions may come across also as rather subjective and a bit slogan like.*

RESPONSE: The words “hitherto unknown” have been deleted from the abstract, and the sentence: “Our findings suggest a previously undiscovered family of topological edge states exists” has been changed to: “Thus, our findings not only extend the way topological edge states are identified, but also promote ...”

2. *The authors claim that the hidden chiral and particle-hole symmetries are related to the coordinate transformations and boundary conditions in their topological system. The authors should clarify more on this issue in the text and incorporate some corresponding parts of the Supplementary Material (SM) into the main text. This is a crucial point that needs to be emphasized.*

RESPONSE: We thank the reviewer for this suggestion. To clarify more the part of the hidden symmetries and how these are related to a suitable choice of coordinates, we have incorporated more details into the main text. For example, in the section of the first case (mass dimer), we have rearranged and modified (incorporating parts of the SM) the discussion regarding the absence of chiral symmetry of the displacement dynamical matrix.

For the second case (mechanical Kitaev chain), the details of bulk dynamical matrices are already included in the Methods section.

The following text has been added with respect to the mass dimer:

“ $\hat{D}_{u,\text{bulk}}(q)\hat{u}(q) = \omega^2\hat{u}(q)$, where $\hat{u} = [\sqrt{P}u_A(q), u_B(q)]^T$, $\hat{D}_{u,\text{bulk}}(q)$ is the Bloch dynamical matrix in displacement coordinates of the following form

$$\hat{D}_{u,\text{bulk}}(q) = \frac{1}{(1+P)} \begin{pmatrix} 2 & -\sqrt{P}(1+e^{-iq}) \\ -\sqrt{P}(1+e^{iq}) & 2P \end{pmatrix}, \quad (\text{R1})$$

and $\omega = \Omega/\Omega_0$ the normalized frequency with respect to the mid-gap frequency $\Omega_0^2 = k(1/m_1 + 1/m_2)$. It is well known that the edge states of the mass dimer appear for free edges when the ratio $P := m_1/m_2$ is varied [40,41]. However, their topological nature has been unknown since the dynamical matrix lacks the necessary symmetries for a topological classification. This is evident when $\hat{D}_{u,\text{bulk}}$ is written in terms of the complex Pauli matrices σ_i ($i = x, y, z$), such that $\hat{D}_{u,\text{bulk}} = I + d_x\sigma_x + d_y\sigma_y + d_z\sigma_z$, where $d_x = \sqrt{P}(1 + \cos q)/(1 + P)$, $d_y = \sqrt{P} \sin q/(1 + P)$, and $d_z = (1 - P)/(1 + P)$. The presence of all the σ_i indicates that the $\hat{D}_{u,\text{bulk}}$ (up to a constant shift in the diagonal) does not anti-commute with any of these matrices. This implies the absence of chiral symmetry as in the standard SSH model [45,46].”

3. I wonder whether the topological elastic waves system has defect immunity. For instance, I wonder if the different boundary conditions could affect the robustness of topologically protected edge states. Can the authors provide corresponding verification?

RESPONSE: We thank the referee for this very valuable question. Topological elastic wave systems have varying degrees of robustness against different types of disorders. We have verified this for both systems (mass dimer and Kitaev mechanical chain) and using different boundary conditions (fixed or free). In Supplementary Note 3 of the revised manuscript, we have added the disorder analysis in detail. From this analysis, we conclude that the edge states appearing in strain metamaterials (for free boundary conditions) have the same level of protection against disorder as in the usual displacement metamaterials (for fixed boundary conditions), for a given type of disorder. To highlight this very important conclusion, we added the following sentence in the introduction: “In addition, such systems have the same level of protection against disorder as the usual passive, finite-frequency topological metamaterials.”

In addition, the following text has been added in Supplementary Note 3:

III. Supplementary Note 3: Disorder analysis

Another implication of topology is the robustness of edge states against disorder. Especially for symmetry-preserving disorders in the chain, the topological edge states are extremely robust. We will provide here evidence that finite-frequency topological mechanical metamaterials, both normal and strain, support edge states that have the same level of protection against various kinds of disorders. For the case of strain topological metamaterials, the disorder analysis is applied in strain coordinates, where, in the corresponding strain dynamical matrices, the symmetry-preserving disorder becomes apparent.

A. Mass vs. stiffness dimer

First, to check the robustness of the edge state in the mass dimer, we present a generalized spring-mass system that could be reduced to either the mass dimer or the stiffness dimer (finite-frequency mechanical SSH chain). The *displacement* dynamical matrix for an $2N$ -particle chain, *fixed* at both ends, has the following tridiagonal form:

$$\hat{D}_u = \begin{bmatrix} \frac{k_1+k_2}{m_1} & -\frac{k_2}{\sqrt{m_1}\sqrt{m_2}} & \dots & 0 & 0 \\ -\frac{k_2}{\sqrt{m_1}\sqrt{m_2}} & \frac{k_2+k_3}{m_2} & -\frac{k_3}{\sqrt{m_2}\sqrt{m_3}} & \dots & 0 \\ \dots & \dots & \dots & \dots & \dots \\ 0 & \dots & -\frac{k_{2N-1}}{\sqrt{m_{2N-2}}\sqrt{m_{2N-1}}} & \frac{k_{2N-1}+k_{2N}}{m_{2N-1}} & -\frac{k_{2N}}{\sqrt{m_{2N-1}}\sqrt{m_{2N}}} \\ 0 & 0 & \dots & -\frac{k_{2N}}{\sqrt{m_{2N-1}}\sqrt{m_{2N}}} & \frac{k_{2N}+k_{2N+1}}{m_{2N}} \end{bmatrix}_{2N \times 2N}. \quad (\text{R2})$$

Similarly, the *strain* dynamical matrix for an $(2N + 1)$ -particle chain, *free* at both ends (i.e., $k_1 = k_{2N+2} = 0$), has the following tridiagonal form:

$$\hat{D}_s = \begin{bmatrix} k_2\left(\frac{1}{m_1} + \frac{1}{m_2}\right) & -\frac{\sqrt{k_2}\sqrt{k_3}}{m_2} & \dots & 0 & 0 \\ -\frac{\sqrt{k_2}\sqrt{k_3}}{m_2} & k_3\left(\frac{1}{m_2} + \frac{1}{m_3}\right) & -\frac{\sqrt{k_3}\sqrt{k_4}}{m_3} & \dots & 0 \\ \dots & \dots & \dots & \dots & \dots \\ 0 & \dots & -\frac{\sqrt{k_{2N-1}}\sqrt{k_{2N}}}{m_{2N-1}} & k_{2N}\left(\frac{1}{m_{2N-1}} + \frac{1}{m_{2N}}\right) & -\frac{\sqrt{k_{2N}}\sqrt{k_{2N+1}}}{m_{2N}} \\ 0 & 0 & \dots & -\frac{\sqrt{k_{2N}}\sqrt{k_{2N+1}}}{m_{2N}} & k_{2N+1}\left(\frac{1}{m_{2N}} + \frac{1}{m_{2N+1}}\right) \end{bmatrix}_{2N \times 2N}. \quad (\text{R3})$$

We start with the classical case of the stiffness dimer chain, and several types of disorders, and compare their behavior to that in the mass dimer. It is easy to see that the displacement dynamical matrix in Eq. (R2) represents the stiffness dimer when $m_i = m$ for $i \in [1, 2N]$, and $k_{2i-1} = k_1 = k_{2N+1}$, $k_{2i} = k_2$ for $i \in [1, N]$. The reduced matrix, therefore, has a constant diagonal ($\frac{k_1+k_2}{m}$), which can be removed, through subtraction of a multiple of the identity matrix, and the remaining matrix is chiral. Similarly, the strain dynamical matrix in Eq. (R3) represents the mass dimer when $k_i = k$ for $i \in [1, 2N]$, and $m_{2i-1} = m_1 = m_{2N+1}$, $m_{2i} = m_2$ for $i \in [1, N]$. Again, the reduced matrix has a constant diagonal [$k(\frac{1}{m_1} + \frac{1}{m_2})$], which can be removed as in the previous case, and the remaining matrix is chiral. We are now ready to introduce several types of disorders in the mass and stiffness dimer chains noting that the effect of disorder on the diagonal terms would be crucial in the topological protection of edge states.

Disorder with no symmetry: In stiffness dimer, we introduce random disorder such that the stiffnesses in the n th unit cell change to $k_{1,n} = k_1 + \gamma\lambda_n$ and $k_{2,n} = k_2 + \gamma\phi_n$, where γ is the disorder strength and λ_n and ϕ_n are random numbers that follow a uniform distribution in $[-1, 1]$. Similarly, in the mass dimer, we vary masses as $m_{1,n} = m_1 + \gamma\lambda_n$, $m_{2,n} = m_2 + \gamma\phi_n$. The initial masses (of the clean case) were chosen such that $P = \frac{m_1}{m_2} = 0.4$, as the experimental sample of mass dimer. For the stiffness dimer we use $\eta = \frac{k_2}{k_1} = 0.4$. In experimentally realized mechanical systems, such as those we present in this work, the mass disorder appears due to modification of the size or density of the particles, while stiffness disorder occurs due to modifications of the beams connecting the particles. In this disorder analysis, we ensure that the disorder strength is experimentally achievable, and we also ensure the stability of the systems by considering only positive quantities for the stiffnesses and masses ($\gamma < 1$). In Supplementary Fig. R1(a), we show the effect of this disorder on the spectrum of stiffness and mass dimers. We present the averaged spectrum of 1000 realizations of disorder for 50 and 51 particle-chains in the case of the stiffness and mass dimers, respectively. We observe the spectra of both the stiffness and mass dimers lose their symmetry about the mid-gap frequency ($\omega^2 = 1$). Particularly, the edge states (localized on both ends) inside the band gap deviate from the midgap frequency. This is because of the loss of chiral symmetry of the finite dynamical matrix after introducing disorder. For the stiffness and mass dimers, the diagonal terms in Eq. (R2) and Eq. (R3) are no longer constant under such disorder, and therefore, it is not possible to restore chiral symmetry after subtraction of a multiple of the identity matrix.

Disorder with inversion symmetry: Next, we introduce disorder in mass and stiffness dimer chains such that the respective disorders are symmetric about the center of the finite chains. In Supplementary Fig. R1(b), we show the averaged spectrum for both stiffness and mass dimer under this disorder. Again, we observe that the spectrum loses its chirality due to this disorder. However, we note the partial protection of the edge states in *both* the mass and stiffness dimers. The two edge states, localized at opposite ends, are not exactly pinned at the mid-gap frequency, but they don't split due to the inversion symmetry of disorder in the finite chains.

Disorder with chiral symmetry: What disorder keeps the chiral symmetry of the stiffness and mass dimer chain? We consider the stiffness dimer chain first. There are two ways to introduce chiral disorder in this chain. The first way is through introducing disorder solely on stiffness, and the second way is also to alter the masses. The first way was introduced recently [5], where the chiral symmetry of the dynamical matrix was preserved by introducing a ground stiffness for each mass. This leads to the addition of an onsite stiffness in the diagonal elements of Eq. (R2). Disorder in k can then be balanced by the disorder in onsite stiffness, to keep the diagonal constant and the matrix chiral.

We present the second way here based on engineered variations of the masses avoiding the introduction of ground stiffnesses. It is clear from Eq. (R2) that the diagonal elements $\eta = \frac{k_i+k_{i+1}}{m_i}$ can be kept constant if the disorder in k terms is balanced by disorder in m terms. We allow $m_i = m + \rho_i$, where m is the initial mass (same for all the particles in the case of the stiffness dimer) and fine-tune ρ_i , such that: $\rho_i = \frac{k_i+k_{i+1}}{\eta} - m$. The same procedure can then be translated to the strain dynamical matrix for the mass dimer in Eq. (R3) as well. We keep the diagonal elements constant, such that $P = k_i(\frac{1}{m_{i-1}} + \frac{1}{m_i})$ by varying $k_i = k + \tau_i$ with $\tau_i = P/(\frac{1}{m_{i-1}} + \frac{1}{m_i}) - k$, where k is the initial stiffness. In Supplementary Fig. R1(c), we show the spectra of mass and stiffness dimer under chiral disorder.

It is evident that this disorder keeps the symmetry of the spectrum about the midgap frequency. The edge states are extremely robust under such disorder. We have also verified that the chiral signature is present in the edge states where their amplitudes vanish at alternating sites (bonds) for the stiffness dimer (mass dimer) when $\gamma = 1$.

Therefore, we have demonstrated that the mass dimer (strain topological chain) supports edge states that are topologically protected against chiral disorder, while they also keep a high level of protection under disorder that respects the inversion symmetry. This provides extra evidence that strain topological metamaterials *are* topological, and their topology (bulk index and protecting symmetries) can only be uncovered in strain coordinates.

Figure R1: **Disorder analysis of the stiffness and mass dimer chains.** Spectra of the stiffness dimer (top row) and mass dimer (bottom row) as a function of the disorder strength γ when (a) no symmetry, (b) inversion symmetry, and (c) chiral symmetry are preserved. Left and right localized edge states are denoted in red and blue.

B. Mechanical Kitaev chain

The mechanical Kitaev chain supports topological edge states at both fixed and free ends. However, the topological origin of the latter is revealed in the strain coordinates. In this section, we will show the edge states for fixed-fixed and free-free chains have equivalent robustness against various types of disorders.

The equations of motion for the mechanical Kitaev chain are:

$$\begin{aligned} \ddot{u}_n &= \frac{K_{S,n}}{m_n} (u_{n-1} - u_n) - \frac{K_{S,n+1}}{m_n} (u_n - u_{n+1}) \\ &+ \frac{K_{S,n}}{m_n} (\Phi_n + \Phi_{n-1}) - \frac{K_{S,n+1}}{m_n} (\Phi_n + \Phi_{n+1}), \end{aligned} \quad (\text{R4a})$$

$$\begin{aligned} \ddot{\Phi}_n &= P_n \frac{K_{S,n}}{m_n} (u_n - u_{n-1} - (\Phi_{n-1} + \Phi_n)) \\ &+ P_n \frac{K_{S,n+1}}{m_n} (u_{n+1} - u_n - (\Phi_n + \Phi_{n+1})) \\ &+ P_n \frac{K_{B,n}}{m_n} (\Phi_{n-1} - \Phi_n) - P_n \frac{K_{B,n+1}}{m_n} (\Phi_n - \Phi_{n+1}), \end{aligned} \quad (\text{R4b})$$

where $P_n = \frac{m_n d_n^2}{I_n}$. We introduce the following types of disorders:

Disorder with no symmetry: We implement disorder without symmetry in two ways. First, we apply a disorder in the mass/inertia (leading to a disorder in values of P_n), and second, apply a disorder in K_B (leading to a disorder in the values of $\eta_n = K_{B,n}/K_S$). The first type of disorder is implemented by inducing a disorder in the height of each particle (H_n), while the second disorder is created by changing the distance between the two beams (s_n). The disorders are constrained to realistic physical quantities (ensuring a possible construction of these structures). The initial conditions, are: $N_{fixed} = 30$ particles with $P = 1.5$, and $N_{free} = 31$ particles with $P \rightarrow P/(P-1)$. We have applied disorder on the mechanical beam in the following manner: $s_n = s_s + s_s \gamma \lambda_n$, where s_s is the starting value without disorder of the distance between the two beams, γ is the disorder strength, and λ_n is a uniform random number in $[-1, 1]$. In an analogous way, the particle height is modified such as $H_n = H_s + H_s \gamma \lambda_n$, where H_s is the starting value without disorder of the particle height. Spectra of the mechanical chain as a function of disorder strength are presented in Supplementary Fig. R2. In Supplementary Figs. R2(a,c), we observe that the edge states are no longer ‘‘pinned’’ at the mid-gap frequency, and they ‘‘split’’ as the disorder strength grows. This is expected due to a random disorder in P and η . However, the degree of protection is seemingly close for the edge states in fixed-fixed and free-free chains.

Disorder with inversion symmetry: We again introduce disorder separately in P and η as before, but now keep the inversion symmetry of the finite disordered chain intact. In Supplementary Figs. R2(b,d), the edge states are now partially protected as they do not split. We observe that inversion symmetry adds an ‘‘extra’’ protection to both fixed-fixed and free-free boundary conditions equally.

Disorder with particle-hole symmetry: Since our mechanical Kitaev chain possess particle-hole symmetry after fine-tuning (as shown in the main text), we look to introduce disorder in this system that preserves this symmetry. Note that since the dynamical matrices are real, particle-hole and chiral symmetry coincide. However, there is no physical way to apply such a disorder in the current mechanical design, without adding additional elements. This is because we have already imposed a constraint ($\eta = 1 - 1/P$) between effective stiffnesses and masses for fine-tuning. Thus we do not have the same freedom for extra fine-tuning to preserve particle-hole as in the case of the mass-stiffness dimer. Since there exists no disorder that preserves the particle-hole (equivalently chiral) symmetry of the fixed-fixed Kitaev chain (in displacement coordinates) in the current mechanical design, it also implies the absence of such disorder in the free-free chain (in strain coordinates). However, as suggested, there could be other ways to introduce disorder that preserves particle-hole symmetry, such as modifying the current mechanical design by adding extra elements, as was done for the stiffness dimer in [5].

Therefore, we have demonstrated that the mechanical Kitaev chain (under the current mechanical design constraints) supports topological edge states that are robust against disorders that respect inversion symmetry. Importantly, we have shown that the edge states at free ends (revealed in strain coordinates) have a similar degree of protection as compared to the edge state at fixed ends (revealed in displacement coordinates).

Figure R2: **Disorder analysis - Kitaev chain.** Spectra of the mechanical Kitaev chain as a function of the disorder strength γ for (a, b) disorder on P (mass/inertia) or (c, d) disorder on η (stiffness). Top line and bottom line refer to fixed-fixed and free-free boundary conditions, respectively. Panels (a, c) display the results when no symmetry is present, while panels (b, d) show the results with inversion symmetry conserved. The insets correspond to a focus around the topological edge state’s frequency. The results are averaged over 1000 iterations for each disorder strength.

4. *The full names of these abbreviations “BDI”, “ODEs” and “DOFs” should be given at the first use.*

RESPONSE: We apologize for this lack of clarity, and we have modified the main text accordingly, giving the full names or descriptions when first used. Here is the summary:

- BDI: a topological class (not an abbreviation) based on one of Cartan’s symmetric spaces, with all time-reversal, chiral, and particle-hole symmetries. Please note that BDI is a label denoting a specific type of Lie Algebra.
- ODEs: ordinary differential equations
- DOFs: degrees of freedom

5. *It is suggested that the authors make a summary at the end of the result section of the text. This can help general readers appreciate the importance of their work.*

RESPONSE: We thank the reviewer for this suggestion. Here is the summary that we have added at the end of the main text: “In summary, we demonstrate theoretically and experimentally herein the existence of a new family of mechanical topological metamaterials in which bulk-boundary correspondence is realized through higher-order coordinates (e.g., strain) and suitable boundary conditions. Such topological states are also robust against several types of disorder in the system, as shown in Supplementary Note 3. We suggest that this is a general finding beyond mechanical metamaterials, as our framework can be applied to any physical system that maps to “mass-spring” systems, including, e.g., electrical or superconducting circuits, optics, and acoustics, among others. In Supplementary Note 4, we provide a supporting example via a photonic setting. Our approach enables the detection

of topological edge states for boundary conditions which at first glance break topology protecting symmetries. Furthermore, it paves the way to design topological metamaterials exploiting the interplay between boundary conditions and higher order coordinates.”

Reviewer #2

In this work, the authors propose an interesting family of mechanical systems named “strain topological metamaterials”. These metamaterials support topological edge states protected by hidden symmetries, that is, symmetries that are not explicitly presented in the original Hamiltonian. The authors show two models (the mass dimer and the mechanical Kitaev models) with such intriguing behavior and address the important role of boundary conditions in preserving the symmetry. Overall, I think this work is of high quality and its results could be impactful in the whole topological classical wave community. I think that this article is suitable for Nature Communications, provided that the following issues could be satisfactorily addressed.

RESPONSE: We thank the reviewer for appreciating the quality and impact of this work.

1. The authors use the name “strain topological metamaterials” in the title. However, it is not clear what the physical meaning of “strain” is, and it seems that D_s and D_u are just related by a basis transformation. Usually, people would expect that strain will change the positions of atomic sites (but not the basis), thus modifying the coupling terms in a Hamiltonian. But here it looks that the on-site terms are also changed. It would be beneficial if the authors can plot the original and strain coordinates and visualize how strain is applied.

RESPONSE: We thank the Reviewer for this question. We note that a strain (static pre-load) is never directly *applied* to our spring-mass systems. We rather interpret our system’s dynamics using higher-order coordinates that correspond to bond extensions (strains) rather than the usual coordinates (displacements of the masses). Indeed, as the referee commented, the bulk dynamical matrices D_s and D_u are related by a basis-like transformation, expressed through the compatibility matrix. At the same time, this transformation has a physical meaning, such that strain is the relative displacement between particles. In the original manuscript, we defined strain as follows: “Given a linear compatibility matrix C defined as $\mathbf{s} = C\mathbf{u}$ [2], we derive equations of motion in terms of bond extensions (strains) $\tilde{\mathbf{s}} = -D_s\mathbf{s}$, where D_s is the strain dynamical matrix.”

Finally, regarding plotting the original and strain coordinates, this comparison was provided in the original version of the manuscript (and is retained in the revised version). In Fig. 2(d), we compare the edge mode profiles for the mass dimer system in displacement (left) and strain (right) coordinates, and in Fig. 4(c,d), we compare the edge mode profiles for the Kitaev chain in both displacement (d) and strain (c) coordinates.

2. There is another type of hidden symmetry (also call latent symmetry) that refers to the symmetry of a sub/reduced part of a system (like in Ref. [26]). I think the authors should mention this to avoid potential conceptual misleading.

RESPONSE: We thank the reviewer for his/her remark. In the introduction, we have rewritten the paragraph discussing hidden symmetries in extended systems clarifying the difference between hidden symmetries and local latent symmetries and adapting appropriately the corresponding references. Here is the added part: “Hidden symmetries have been widely shown to exist in virtually every branch of physics [23–26]. They can be revealed after mathematical mappings, suitable coordinate transformation [27–31], or by isospectral reductions [32–33] the so-called latent symmetries”

3. An important yet unaddressed question is how the hidden symmetry can protect the topological states in the real system. Using the chiral symmetry as an example. For a conventional topological

system with chiral symmetry, symmetry-preserving disordering (e.g., disorder in the couplings) cannot move the topological edge states from zero energy. However, for the hidden chiral symmetry, it is not clear how it can be preserved upon perturbation/disorder to the real system. From Eq. (3), it seems that any perturbation/disorder in P will destroy the chiral symmetry.

RESPONSE: We thank the reviewer for this excellent remark. As per the question asked by Reviewer 1 as well, in Supplementary Note 3 of the revised version, we have added a detailed disorder analysis for both the displacement and strain topological metamaterials.

For the class of passive, finite frequency topological mechanical metamaterials, any disorder in the stiffnesses (couplings) does break the chiral symmetry of the system. This is a fundamental difference between such mechanical lattices and the conventional, tight-binding topological lattices, for which disorder in the couplings (occurring only in off-diagonal elements of the Hamiltonian) does not break the chiral symmetry. It is the disorder on the on-site energies (occurring in diagonal of the Hamiltonian) that break the chiral symmetry in such systems.

However, as we show in detail in the SM note, one can keep the chiral symmetry of mechanical systems as well, by an engineered disorder (simultaneously in the masses and the stiffnesses). In usual displacement, topological metamaterials, e.g., a stiffness dimer, a disorder in stiffnesses, can be balanced by an engineered disorder in masses. In the same way, in strain topological metamaterials, e.g., a mass dimer, a disorder in masses can be balanced by an engineered disorder in stiffnesses. However, in the latter the chirality condition is apparent only when we express the dynamical matrix in strain coordinates. Consequently, the edge states in the mass dimer are pinned to a finite frequency for a chiral disorder.

Overall, we have shown that the edge states (typical and strain topological metamaterials) have the same degree of robustness against several types of disorders. This provides further verification that strain topological metamaterials are indeed a new family of topological metamaterials.

4. There are two models in the paper. But only one of them is experimentally demonstrated (i.e., the Kitaev model). In my opinion, the mass dimer model is the minimal model to demonstrate the authors' idea (and it is easier to realize in experiments). Thus, I suggest the authors also put some experimental results in Fig. 2.

RESPONSE: We thank the reviewer again for their valuable suggestion. We have performed new experiments on the mass dimer to further demonstrate the concept presented herein. We have included these new results in the “Experimental results” section of the revised main text. Figure 5 has been revised with the experimental results of the mass dimer. Additional details are in Methods and Supplementary Note 1.

5. How is the quantity “localization” calculated? (e.g., in Fig. 2(b))

RESPONSE: We have added a new paragraph in the revised Methods section which explains of definition and calculation of localization, as follows: “Figures 2(b), 3(e), and 4(b) display a colormap (from yellow to blue) which represents a localization index. For an eigenvector with N masses, we use the localization index (i.e., inverse participation ratio) defined as:

$$\text{IPR} = \frac{\sum_{n=1}^N A_n^2}{\left(\sum_{n=1}^N A_n^2\right)^2}, \quad (\text{R5})$$

where $A_n = \sqrt{u_n^2}$ or $A_n = \sqrt{u_n^2 + \frac{\Phi_n^2}{P}}$ for the mass dimer and Kitaev analyses, respectively. The inverse participation ratio (IPR) is equal to zero (yellow color) when all particles are moving (delocalized mode) and is near to one (blue color) when only a few particles are moving compared to the total length of the chain as in the case of a spatially localized mode.”

6. It would be good for understanding if a colormap can be added to the yellow curve in Fig. 3(b)

to show the values of P on the curve.

RESPONSE: We thank the reviewer for this suggestion. A colormap has been added to Fig. 3(b) as suggested. The revised panel is as in Fig. R3.

Figure R3: New panel (b) of Fig. 3 from the main manuscript.

Reviewer #3

The authors propose the concept of “strain topological metamaterials”, in which the topology is revealed only in strain coordinates, and the edge states can exist only for free boundaries. The concept proposed in the paper is interesting. However, I cannot recommend the publication of the work for several reasons:

- 1. The paper is too technical. I found it more suitable for a physics journal like Physical Reviews. The focus on the experimental side is very little, restricted to a the very specialized case of mass spring system. If the concept was demonstrated in photonics, I would likely recommend the publication of the work. However, in the current form, the real applicability of the proposed theory is questionable to me. The authors state that the second order ODEs, such as those describing spring-mass models, are universal models for all fields of wave physics, from mechanical to optical [33], acoustical [34] systems. This statement, however, is not enough in my eye to claim that the proposed concept is easily extendable to optics or acoustics. That being said, as long as the experimental realization of the model is limited to this simple mass spring experiment, I cannot recommend the publication of the work.*

RESPONSE: We understand that the reviewer has two main concerns here. The first concern is about the limited focus on experiments. In the revised manuscript, we have included new experiments on the mass dimer to demonstrate the applicability of our idea further. We included the results in the “Experimental results” section of the main text. Figure 5 has been revised with the experimental results of the mass dimer. Additional details are in Methods and Supplementary Note 1.

The second concern is about the general applicability of mass-spring models. As cited in our original manuscript, various physical systems can be mapped (in a targeted range of frequencies) to mass-spring models. For example, this is the case for electrical circuits [3], acoustics [1], phononic/photonic crystals [6] or photonic networks[4]. In particular, in Ref. [6], the authors demonstrated that phononic and photonic crystals consisting of closely spaced inclusions, are asymptotically equivalent at low frequencies to mass-spring models. In the revised version, in order to further demonstrate the applicability of our proposed concepts, and in particular in photonics, we have added a new Supplementary Note 4, where we use full-field finite element method (FEM) simulations to show that photonic systems can be indeed mapped to mass-spring models. To achieve this, we follow a mapping previously

demonstrated in Ref. [4]. We repeat the demonstration from Ref. [4] of a photonic system that follows the dynamics of the topologically protected SSH mass-spring chain, and add a new photonic system example that maps to the mass dimer system shown in our work. Below is the added supplementary information for the Reviewer's convenience:

IV. Supplementary Note 4: Photonic analogies to mass and spring models

Here we show a class of photonic systems for which the transverse-electric polarised light dynamics can map to those of a mass and spring model within certain parameter regimes, as demonstrated in Ref. [4]. The photonic system is composed of a series of voids of half height $H_{1,2}$ connected by thin channels of half height $h_{1,2}$, as shown in Fig. R4(a). Each unit cell for the dimer systems is composed of two voids and two channels and has unit cell spacing $a = 2L$. The boundaries of the voids are composed of a perfect electrical conductor and the areas are vacuum. As shown in [4] the mapped masses are proportional to the area of the voids, such that:

$$m_i = A_i \cdot m_0 / L^2, \quad (\text{R6})$$

where A_i is the area of the i -th void and m_0 is defined in terms of the speed of light c_0 such that $c_0 = \sqrt{k_0/m_0}$. The mapped spring constants are defined as:

$$k_i = \frac{1}{\pi} \sqrt{\frac{2h_i}{R_i}} \cdot k_0, \quad (\text{R7})$$

where R_i is the radius of curvature of the side of the channel and k_0 is defined similarly to m_0 in terms of c_0 .

Figure R4: Photonic SSH mass and spring chain analog. (a) Geometry and relevant parameters. (b) Spectrum normalized to the analytical center frequency $\omega_m^2 = (k_1 + k_2)/m$ calculated using both FEM and DEM. (c) The topological edge mode visualized in terms of the out of plane magnetic field H_z . A topological edge mode (mode index 10) calculated both using FEM and DEM. (d) A topological edge mode (mode index 10) calculated both using FEM and DEM.

A. Photonic Su-Schrieffer-Heeger (SSH) chain

A photonic SSH chain is modeled by setting the heights of each void to be the same ($H_1 = H_2 = H$). For a topological SSH chain the intra-cell coupling needs to be greater than the inter-cell hopping, *i.e.* $k_2 > k_1$. Here, we use same parameters as in Ref. [4] ($H = L/10$, $h_1 = H/400$, $h_2 = H/100$) and set $L = 5 \mu\text{m}$ to create a $N = 20$ particle chain. Both ends of the chain are capped with circular voids of radius $L/2$ to provide “fixed” boundary conditions. The geometry of the chain is given by the function:

$$y(x) = \pm[H \sin(\pi x/L)^2 + h_1 \sin(\pi x/(2L))^2 + h_2 \cos(\pi x/(2L))^2], \quad (\text{R8})$$

where $x \in [-L/4, (N + 1/4)L]$. For this case, the $R_i = R = L^2/(2\pi^2 H)$ and $A_i = A = HL$, such that the bulk of the chain is composed of identical masses m and alternating springs with stiffness k_1 and k_2 . The left and right end caps are given respectively by:

$$X_L(\theta) = \frac{L}{2} \cos(\theta) - \frac{3}{4}L, \quad (\text{R9})$$

$$Y_L(\theta) = \frac{L}{2} \sin(\theta) + h_{end} \cos(\theta/2), \quad (\text{R10})$$

$$X_R(\theta) = \frac{L}{2} \cos(\theta) + (N + \frac{3}{4})L, \quad (\text{R11})$$

$$Y_R(\theta) = \frac{L}{2} \sin(\theta) - h_{end} \sin(\theta/2), \quad (\text{R12})$$

with $h_{end} = y(-L/4) = y(N + 1/4)L$. The resulting geometry is shown in Fig. R4(a). The finite spectrum and normal modes are calculated with both FEM using COMSOL Multiphysics v6.0 and a discrete element model (DEM) using MATLAB R2021a, and are shown in Fig. R4(b-d).

B. Photonic mass dimer

A photonic equivalent of a mass dimer system is modeled by setting $p = m_1/m_2 = 0.25$. We set the first mass to have height $H_1 = L/20$, and the second height as a scalar multiple times the first height $H_2 = \alpha H_1$. The chain geometry is given by the function:

$$y(x) = \pm[((H_1 + H_2)/2 + ((H_1 - H_2)/2) \sin(\pi x/L)) \sin(\pi x/L)^2 + h], \quad (\text{R13})$$

noting that L is set to be the same as in the SSH example and $h = h_1 = h_2 = H_1/400$. The height scaling factor α is found by integrating Eq. R13 to achieve the desired mass ratio, such that $\alpha = 5.8226$. This results in a chain where the bulk of the chain is composed of alternating masses m_1 and m_2 and identical springs with stiffness $k = k_1 = k_2$, as defined in Eq. R7. For calculating the stiffness k , as in Eq. R7, we define the radius of curvature of the channel to be:

$$R = \left| \frac{(1 + \frac{dy}{dx})^{3/2}}{\frac{d^2y}{dx^2}} \right| = \frac{L^2}{\pi^2(H_1 + H_2)}, \quad (\text{R14})$$

where $y(x)$ is given by Eq. R13. Free boundary conditions are implemented by cutting the geometry at the edge of a unit cell and not including end caps. The system parameters were chosen so that it maps to the mass dimer system considered in the experimental section of the main text for $P = 0.25$, $N = 7$. The resulting geometry is shown in Fig. R5(a). The finite spectrum and normal modes are again calculated with both FEM and DEM, shown in Fig. R5(b-d). ”

2. The authors also do not show the applicability of the proposed concept in higher dimensions. A typical TIs is often used for robust transportation of waves. But I am afraid that the proposed concept will lead to a novel form of topologically protected transport. Can the authors show the robust transport of energy in the strained coordinates? In my opinion, it is not feasible. If it is true, the applicability of the proposed concept is mitigated more.

Figure R5: Photonic mass dimer analog equivalent to the experimental system studied in the main text. **(a)** Geometry. **(b)** Spectrum calculated through both DEM and FEM for the photonic mass dimer normalized by $\omega_m^2 = k(1/m_1 + 1/m_2)$. **(c)** Topological edge mode for mass dimer system (mode index 7). **(d)** Calculated topological edge modes from both FEM and DEM (mode index 7) and normalized to the largest amplitude field component H_z .

RESPONSE: Indeed, applying the idea of hidden topological symmetries in two-dimensional settings is a natural extension of our work. This is currently under investigation and the results will be presented in a separate future publication. However, we have included a detailed disorder analysis in Supplementary Note 3 showing that the localized edge states in 1D topological metamaterials possess the same degree of robustness in displacement and strain coordinates.

References

- [1] V. Achilleos, G. Theocharis, O. Richoux, and V. Pagneux. Non-hermitian acoustic metamaterials: Role of exceptional points in sound absorption. *Phys. Rev. B*, 95:144303, Apr 2017.
- [2] C. Kane and T. Lubensky. Topological boundary modes in isostatic lattices. *Nat. Phys.*, 10(1):39–45, 2014.
- [3] C. H. Lee, S. Imhof, C. Berger, F. Bayer, J. Brehm, L. W. Molenkamp, T. Kiessling, and R. Thomale. Topoelectrical circuits. *Commun. Phys.*, 1:39, 2018.
- [4] S. Palmer, Y. Ignatov, R. Craster, and M. Makwana. Asymptotically exact photonic approximations of chiral symmetric topological tight-binding models. *New Journal of Physics*, 24(5):053020, 2022.
- [5] X. Shi, I. Kiorpelidis, R. Chaunsali, V. Achilleos, G. Theocharis, and J. Yang. Disorder-induced topological phase transition in a one-dimensional mechanical system. *Phys. Rev. Research*, 3:033012, Jul 2021.
- [6] A. L. Vanel, O. Schnitzer, and R. V. Craster. Asymptotic network models of subwavelength metamaterials formed by closely packed photonic and phononic crystals. *Europhysics Letters*, 119(6):64002, nov 2017.

REVIEWERS' COMMENTS

Reviewer #1 (Remarks to the Author):

In the revised manuscript, the authors addressed all my comments. Now the current manuscript is much improved and the key information become complete. At this level I would like to recommend its publication.

Reviewer #2 (Remarks to the Author):

I thank the authors for properly addressing my concerns. There is a typo in Eq. (23) in the main text (the power of the numerator should be 4). After fixing this, I think the manuscript can be accepted for publication.

Reviewer #3 (Remarks to the Author):

The authors have addressed my comments appropriately. I was particularly fascinated by the demonstration of the concept in a photonic structure, which has been added to the revised version of the manuscript. This demonstration addresses my major concern regarding the applicability of the concept in other fields. Given this newly added study, I have no objection against publication of the work.

Responses to the reports of Referees – NCOMMS-23-06225B

Reviewer #1

In the revised manuscript, the authors addressed all my comments. Now the current manuscript is much improved and the key information become complete. At this level I would like to recommend its publication.

Reviewer #2

I thank the authors for properly addressing my concerns. There is a typo in Eq. (23) in the main text (the power of the numerator should be 4). After fixing this, I think the manuscript can be accepted for publication.

Reviewer #3

The authors have addressed my comments appropriately. I was particularly fascinated by the demonstration of the concept in a photonic structure, which has been added to the revised version of the manuscript. This demonstration addresses my major concern regarding the applicability of the concept in other fields. Given this newly added study, I have no objection against publication of the work.

RESPONSE: We thank the reviewers for appreciating the quality and impact of this work. We have corrected the typo in Eq. (23).